# Fasoracetam in adolescents with ADHD and glutamatergic gene network variants disrupting mGluR neurotransmitter signaling

Josephine Elia[1,2,3], Grace Ungal[4], Charlly Kao[5], Alexander Ambrosini[6], Nilsa De Jesus-Rosario[3], Lene Larsen[3], Rosetta Chiavacci[5], Tiancheng Wang[5], Christine Kurian[2], Kanani Titchen[1,2], Brian Sykes[1,2], Sharon Hwang[1,2], Bhumi Kumar[1,2], Jacqueline Potts[2], Joshua Davis[1,2], Jeffrey Malatack[1], Emma Slattery[2], Ganesh Moorthy[7], Athena Zuppa[7], Andrew Weller[3], Enda Byrne[5], Yun R. Li[5,8], Walter K. Kraft [2] & Hakon Hakonarson[5,7,9]

The glutamatergic neurotransmitter system may play an important role in attention-deficit hyperactivity disorder (ADHD). This 5-week, open-label, single-blind, placebo-controlled study reports the safety, pharmacokinetics and responsiveness of the metabotropic glutamate receptor (mGluR) activator fasoracetam (NFC-1), in 30 adolescents, age 12–17 years with ADHD, harboring mutations in mGluR network genes. Mutation status was double-blinded. A single-dose pharmacokinetic profiling from 50–800 mg was followed by a single-blind placebo at week 1 and subsequent symptom-driven dose advancement up to 400 mg BID for 4 weeks. NFC-1 treatment resulted in significant improvement. Mean Clinical Global Impressions-Improvement (CGI-I) and Severity (CGI-S) scores were, respectively, 3.79 at baseline vs. 2.33 at week 5 ($P < 0.001$) and 4.83 at baseline vs. 3.86 at week 5 ($P < 0.001$). Parental Vanderbilt scores showed significant improvement for subjects with mGluR Tier 1 variants ($P < 0.035$). There were no differences in the incidence of adverse events between placebo week and weeks on active drug. The trial is registered at https://clinicaltrials.gov/ct2/show/study/NCT02286817.

[1] Nemours, du Pont Hospital for Children, Wilmington 19803 DE, USA. [2] Department of Pediatrics Sidney Kimmel Medical College of Thomas Jefferson University, Philadelphia 19107 PA, USA. [3] Department of Psychiatry Sidney Kimmel Medical College of Thomas Jefferson University 19107 Philadelphia, USA. [4] Drexel University College of Medicine, Philadelphia 19129 PA, USA. [5] The Center for Applied Genomics, The Children's Hospital of Philadelphia, Philadelphia 19104 PA, USA. [6] University of Pennsylvania, Philadelphia 19104 PA, USA. [7] Department of Pediatrics Perelman School of Medicine, University of Pennsylvania, Philadelphia 19104 PA, USA. [8] Department of Radiation Oncology, Helen Diller Family Cancer Center, University of California San Francisco, San Francisco 94143 CA, USA. [9] Divisions of Human Genetics and Pulmonary Medicine, The Children's Hospital of Philadelphia, Philadelphia 19104 PA, USA. Correspondence and requests for materials should be addressed to J.E. (email: Josephine.Elia@nemours.org)

Attention-deficit hyperactivity disorder (ADHD) has a prevalence of ~8% in children with a 2:1 ratio of males and females affected[1]. Symptoms persists into adulthood in over two thirds of cases, causing significant life-long impairments[2,3]. While some genetic risk factors for ADHD have been identified, a significant gap exists in the translation of this knowledge to identifying therapeutics with clinical utility. Standard of care medications used to treat ADHD have been largely unchanged for the last few decades and the most effective treatments largely consist of stimulants, which are non-specific and cause significant off-target effects[4,5].

A recent large-scale, genome-wide analysis showed that CNVs affecting a set of nearly 300 genes in the metabotropic glutamatergic network (mGluR) occurred at a significantly higher frequency in children with ADHD[6]. CNVs in these genes were detected in 11.3% of ADHD cases vs. 1.2% of controls[6], based on an analysis of 3500 ADHD cases and nearly 12,000 control subjects[6]. Importantly, a core 79-gene subset was enriched in patients with ADHD group by over 10-fold[6,7].

The glutamatergic system has been shown to play a direct role in several animal models of ADHD[8,9]. For example, deletion of the glutamate receptor metabotropic gene, GRM5, and inhibition of the metabotropic glutamate receptor protein mGluR5 with a pharmacologist antagonist resulted in increased spontaneous locomotor activity in rats[8]. Comparable results were obtained when GRM7[9] and GRM8[10], were knocked out in mice. In human subjects, imaging studies have shown increased glutamatergic signaling in fronto-striatal pathways, in many but not all ADHD youths[11], while decreased signaling has been reported in ADHD adults[11,12]. Glutamate changes in children and adolescents treated with ADHD medication also show mixed results[11], suggesting that there may be a subgroup of ADHD subjects with impaired glutamatergic activity. Altogether, these findings suggest that mutations in the mGluR system occurs in at least a subset of individuals with ADHD, warranting further research to study glutamatergic agents as a targeted therapeutic strategy in ADHD[5,13].

NFC-1 (fasoracetam monohydrate) is a small synthetic molecule and a metabotropic glutamate receptor activator, which has previously undergone extensive Phase I-III clinical trials in humans for vascular dementia[14]. Preclinical studies demonstrated that that NFC-1 affects mGluRs[15–17], acetylcholine release and uptake[18] and GABAb, but not adrenoceptors, serotonergic or dopaminergic receptors[19], potentially reversing learning and memory deficits caused by dysfunction of central cholinergic neurons in a variety of models[15,16]. It may have the potential to restore normal glutamatergic activity in ADHD patients with glutamatergic hypofunction due to mutations in mGluR network genes. Aside from focusing on CNVs in the top 79 genes identified in the main mGluR network (Tier-1 mutations)[5], we included CNVs in all 279 previously defined mGluR network genes (200 belonging to Tier-2) as well as an additional set of 600 genes that come from other related gene networks that interact with the mGluR network genes (Tier-3 genes)[5]. Rare CNVs in these genes were used to identify and prioritize patients who may benefit from NFC-1.

The primary goal of this study was to investigate the safety and pharmacokinetics of the mGluR activator NFC-1 molecule in children with ADHD. A secondary objective was to determine if NFC-1 efficacy in alleviating ADHD symptoms in patients stratified by the presence of germline mutations in glutamatergic network genes as measured by established survey instruments and rating scales[20], including Clinical Global Impression Scales for Improvement and Severity (CGI-I and CGI-S, respectively)[21,22], Vanderbilt Parent Scale (www.nichq.org)[23] and the parental Behavior Rating Inventory of Executive Function (BRIEF)[24,25] as well as by changes in overall activity levels by actigraphy[26].

**Table 1 Demographic information of the study participants**

|  | Caucasian | African American | Hispanic | Row totals | Mean age |
|---|---|---|---|---|---|
| *Males (all)* | 11 | 7 | 2 | 20 | 14.6 yo |
| 12–13 yo | 2 | 3 | 2 | 7 |  |
| 14–15 yo | 4 | 1 | 0 | 5 |  |
| 16–17 yo | 5 | 3 | 0 | 8 |  |
| *Females (all)* | 4 | 6 | 0 | 10 | 14.4 yo |
| 12–13 yo | 1 | 2 | 0 | 3 |  |
| 14–15 yo | 2 | 1 | 0 | 3 |  |
| 16–17 yo | 1 | 3 | 0 | 4 |  |
| *Column totals* | 15 | 13 | 2 | 30 | 14.5 yo |

## Results

We conducted an open-label, single-blind-fixed placebo (week 1), followed by a 4 week dose-escalation study in tandem to a 24-hour PK study (see Supplementary Tables 1 and 2 for study flow chart and study parameters collected at each visit). Out of 30 participants enrolled, 29 completed all study time points; 1 completed all but the last time point. All 30 subjects were included in the analysis. Demographic information on the study subjects is provided in Table 1. NFC-1 was found to be safe and well-tolerated. The PK profile was comparable to profiles previously reported for NFC-1 in adult human subjects, based on analysis of PK parameters $C_{max}$, $T_{max}$, and $AUC_{0–24h}$ (Table 2). PK parameters demonstrated dose linear pharmacokinetics. Reported adverse events (AEs) were generally mild, and non-treatment limiting (Table 3). None of the AEs were associated with study drug use, since no difference was observed in the frequencies of any of the AEs reported between the placebo week (week 1) and any of the active drug weeks (weeks 2–5) (Supplementary Table 3).

There were three serious adverse events (SAEs) during the study, none of which was attributed to the study drug. One subject was hospitalized in a pediatric center for an assessment of head injury just prior to his last appointment. He was the only subject who did not complete the study. A second subject experienced dizziness and brief loss of consciousness (5–10 s) in the middle of the school day after medication dose had been titrated to 100 mg that morning. On evaluation within 2 h, his vital signs and physical exam were normal. The second dose of study drug was held on that day and subsequently decreased to 50 mg (same as previous week). There were no recurrences of dizziness and dose titration resumed again without recurrence of dizziness. A third subject had an elevation of creatinine phosphokinase (CPK) to about ~20,000, which was attributed to intensive and strenuous sports training; NFC-1 was discontinued initially, and resumed upon normalization of CPK Levels, which did not rise again while on NFC-1 and despite ongoing regular exercise.

Primary efficacy was measured by cumulative changes in global rating scales CGI-I, CGI-S and Vanderbilt, and Brief scores. Overall, patients showed significant improvements in all four clinical measures by week 5 of the study, as compared to week 1, during which all patients received placebo (range of $P < 0.05$ to $P < 1.2 \times 10^{-9}$; Table 4). The strongest improvements were noted in the CGI-I score, which fell from a mean (median) baseline score of $3.79(4) \pm 0.81$ (minimally improved to no change), to a mean (median) score of $2.33(2) \pm 0.71$ (moderately to much improved) after receiving 400 mg BID of NFC-1 during week 5 ($P < 1.2 \times 10^{-5}$) (Fig. 1).

We further examined the effects of NFC-1 on study subjects stratified by the presence of specific mGluR variants, though this information was double blinded during the study. A total of

**Table 2 Pharmacokinetic parameters**

| Parameter | Dose | | | | |
|---|---|---|---|---|---|
| | **50 mg** | **100 mg** | **200 mg** | **400 mg** | **800 mg** |
| $T_{max}$ (h) | 1.5 ± 0.9 | 1.9 ± 1.1 | 1.3 ± 0.6 | 1.3 ± 0.6 | 1.9 ± 1.9 |
| $C_{max}$ (µg/ml) | 1.19 ± 0.39 | 1.72 ± 0.59 | 5.07 ± 0.83 | 10.77 ± 2.69 | 20.52 ± 7.21 |
| $AUC_{(0-\infty)}$ (h × µg/ml) | 6.87 ± 1.35 | 15.68 ± 4.97 | 30.20 ± 5.70 | 58.11 ± 12.15 | 136.46 ± 29.22 |
| $T_{1/2}$ (h) | 4.44 ± 0.65 | 6.99 ± 4.72 | 4.48 ± 0.65 | 4.11 ± 0.47 | 4.06 ± 0.47 |

A total of six (6) subjects were included in each study/dose group

17 subjects had genomic deletions or disruptive duplications in Tier-1 mGluR genes, while 7 were categorized as having genomic deletions or disruptive duplications in Tier-2 mGluR genes and 6 in one of the more distantly-related mGluR network genes (Tier 3), which are potentially of less relevance to mGluR signaling. Although CGI-I improvements were noted across all patient groups, among subjects who dose escalated to 400 mg BID, patients in the Tier 1 and Tier 2 groups showed superior response ($P < 3.1 \times 10^{-6}$, $P < 2.1 \times 10^{-3}$, respectively). Comparatively, the responses in Tier 3 subjects being notably less ($P < 0.053$). Final mean CGI-I scores decreased by more than 1.5 points for Tiers 1 and 2, while the mean decreased by 1.2 points in Tier 3 patients, when compared to placebo week (Fig. 2).

A similar trend of improvement for the CGI-S index was observed when comparing the net change from baseline to week 5 across all patients with mean CGI-S scores falling from a mean (median) score of 4.86(5) ± 0.57 (moderately to severely ill), to a mean (median) score of 3.93(4) ± 0.90 (mildly to moderately ill) after receiving 400 mg BID of NFC-1 week 5 ($P < 1.7 \times 10^{-5}$). No subjects demonstrated worsening of CGI-S during the study (Fig. 1).

Importantly, baseline CGI-S scores were not significantly different when compared to CGI-S scores of patients during week 1, during which patients receiving placebo ($P = 0.82$) or NFC-1 at 50 mg BID ($P = 0.18$). However, significant improvement of CGI-S scores were observed after patients received at least 100 mg BID ($P = 0.04$), 200 mg BID ($P = 0.04$), and 400 mg BID ($P < 0.001$) as compared to baseline, suggesting that a minimum of 100 mg of NFC1-1 was required to observe significant therapeutic benefit and that the improvement observed is not likely attributable to placebo.

In addition, as with CGI-I scores, the CGI-S score improvements were stratified by genetic tier, as Tier 1 subjects had the greatest mean reduction in CGI-S scores (Fig. 2). Mean CGI-S scores decreased by at least 1 point in both Tier 1 and Tier 2 subjects ($P < 3.3 \times 10^{-4}$ and $P < 8.5 \times 10^{-3}$), while only a modest 0.3 point absolute reduction in the mean was noted in Tier 3 ($P = 0.45$) (Table 5).

As a whole, patients also demonstrated significant, albeit more moderate improvements in Vanderbilt and Brief indices between study baseline and week 5 ($P < 0.01$ and 0.05, respectively). While patients from across all three tiers showed a trend to improved Vanderbilt scores (Supplementary Fig. 4), only those in Tier 1 showed a statistically significant reduction ($P < 0.035$). For the Brief index, we observed an overall trend to improvement across all tiers, though this was not statistically significant. We also observed a modest, though not statistically significant, improvement in every study sub-measure (Supplementary Fig. 5).

Finally, we also examined the effect of NFC-1 on the over-activity domain of ADHD symptoms exhibited by the study subjects by monitoring children's activity levels by actigraphy[26]. Actigraphy monitoring was performed through the entire study period for each patient and standard measures were assessed[27,28]. There was a net reduction in moderate to high intensity and

**Table 3 Treatment Emergent Adverse Events (TEAEs)**

| Severity | Any | | Mild | | Moderate | |
|---|---|---|---|---|---|---|
| | **n** | **% total** | **n** | **% total** | **n** | **% total** |
| Headache | 19 | 63.3% | 18 | 60.0% | 1 | 3.3% |
| Fatigue | 11 | 36.7% | 9 | 30.0% | 2 | 6.7% |
| Abdominal Pain Upper | 8 | 26.7% | 7 | 23.3% | 1 | 3.3% |
| Diarrhea | 7 | 23.3% | 7 | 23.3% | 0 | 0.0% |
| Irritability | 6 | 20.0% | 5 | 16.7% | 1 | 3.3% |
| Dizziness | 4 | 13.3% | 4 | 13.3% | 0 | 0.0% |
| Pyrexia | 4 | 13.3% | 4 | 13.3% | 0 | 0.0% |
| Anxiety | 3 | 10.0% | 2 | 6.7% | 1 | 3.3% |
| Somnolence | 3 | 10.0% | 2 | 6.7% | 1 | 3.3% |
| Onychophagia | 3 | 10.0% | 2 | 6.7% | 1 | 3.3% |
| Tearfulness | 3 | 10.0% | 3 | 10.0% | 0 | 0.0% |
| Depressed Mood | 3 | 10.0% | 3 | 10.0% | 0 | 0.0% |
| Cough | 3 | 10.0% | 3 | 10.0% | 0 | 0.0% |
| Oropharyngeal Pain | 3 | 10.0% | 3 | 10.0% | 0 | 0.0% |
| Vomiting | 2 | 6.7% | 2 | 6.7% | 0 | 0.0% |
| Memory Impairment | 2 | 6.7% | 2 | 6.7% | 0 | 0.0% |
| Social Avoidant Behavior | 2 | 6.7% | 2 | 6.7% | 0 | 0.0% |
| Visual Impairment | 2 | 6.7% | 2 | 6.7% | 0 | 0.0% |
| Nausea | 2 | 6.7% | 2 | 6.7% | 0 | 0.0% |
| Confusion | 2 | 6.7% | 1 | 3.3% | 1 | 3.3% |

Number and percent of subjects reporting TEAEs occurring in more than 5% of the study population. No difference was observed in the frequencies of any of the adverse events reported between the placebo week (week 1) and any of the active drug weeks (weeks 2–5). Please see Supplementary Table 3 for detailed week by week break down of AEs

repetitive movements (Table 6), as measured by mean Moderate to Vigorous Physical Activity (MVPA) per hour, by week 5 of drug use as compared to that during the week of placebo ($P < 3.5 \times 10^{-4}$). In addition, we observed improvements in several other parameters including mean hourly energy expenditure ($P < 0.01$), percent of time spent in MVPA ($P < 5.3 \times 10^{-3}$), and vector magnitude average counts ($P < 9 \times 10^{-4}$). As the actigraphy measures are objective in nature, they support the notion that the measured efficacy observed in CGI-I, CGI-S and Vanderbilt ranting scores are unlikely to be attributed to placebo effects or other measuring bias. Measures of the two major metabolites yielded levels that were over 2-orders of magnitude lower than for NFC1 parent, suggesting they were non-contributory.

## Discussion

The objectives of this study were to explore the safety, pharmacokinetic parameters and potential efficacy of a glutamate receptor activator NFC-1 in adolescents with ADHD and disruptive mutations in genes impacting the mGluR network. While

**Table 4 Study measure scores for all subjects and stratified by genetic tier**

| | | All | Tier 1 | Tier 2 | Tier 3 |
|---|---|---|---|---|---|
| *CGI-I* | *P*-value | *1.2E−09* | *3.1E−06* | *2.1E−03* | *5.3E−02* |
| | Baseline | 3.79 (4) ± 0.81 | 3.93 (4) ± 0.92 | 3.57 (3) ± 0.78 | 3.66 (4) ± 0.51 |
| | Final | 2.33 (2) ± 0.71 | 2.23 (2) ± 0.75 | 2.14 (2) ± 0.37 | 2.83 (3) ± 0.75 |
| *CGI-S* | *P*-value | *1.7E−05* | *3.3E−04* | *8.5E−03* | *4.5E−01* |
| | Baseline | 4.86 (5) ± 0.57 | 4.88 (5) ± 0.60 | 4.71 (5) ± 0.48 | 5 (5) ± 0.63 |
| | Final | 3.93 (4) ± 0.90 | 3.82 (4) ± 0.88 | 3.57 (3) ± 0.78 | 4.66 (4.5) ± 0.81 |
| *Vanderbilt (P)* | *P*-value | *1.0E−01* | *3.5E−02* | *1.8E−01* | *5.2E−01* |
| | Baseline | 28.7 (32.5) ± 13.9 | 28.7 (32) ± 14.5 | 33.8 (37) ± 12.4 | 22.8 (26) ± 13.7 |
| | Final | 19.7 (17.5) ± 12.2 | 18.5 (17) ± 12.2 | 24.4 (23) ± 12.3 | 17.6 (16.5) ± 12.8 |
| *BRIEF (P)* | *P*-value | *4.9E−02* | *1.8E−01* | *1.9E−01* | *4.7E−01* |
| | Baseline | 68.4 (70.3) ± 11.2 | 67.2 (70) ± 11.2 | 73.8 (71.0) ± 8.85 | 65.9 (71.4) ± 13.5 |
| | Final | 62.1 (63.1) ± 13.0 | 61.3 (61.9) ± 13.7 | 66.0 (63.1) ± 11.7 | 60.0 (63.0) ± 13.5 |

Scores are given in the format of mean (median) ± s.d. Baseline refers to week 1 (fixed single-blind placebo week). Final refers to week 5 where patients received the maximal dose for that study allocation group. *P*-values are calculated using a two-sided, paired Student's *t*-test comparing scores obtained at baseline (week 1) vs. final week of study drug

the heritability for many of the common complex psychiatric or neurodevelopmental diseases, including ADHD, are high, the utility of specific genetic associations as disease biomarkers in guiding the diagnosis or treatment of has been challenging. Our study is the first to use NFC-1 to treat adolescents with ADHD who are carriers of mGluR risk variants, and the first time this compound is used in the United States. This trial demonstrates the feasibility of an innovative, precision medicine based trial design that leverages data from genetic testing to both identify a compound not previously registered and to repurpose it for use in a genetically stratified data set.

We showed that adolescents with ADHD, who are found to have germline mutations impacting the GRM network, show clinical improvement using global improvement and severity scales as well as ADHD symptom scales in response to escalating dosages of NFC-1, a non-selective glutamate receptor activator with additional GABA$_B$ and cholinergic enhancing effects[14]. We observed no significant changes in CGI-S scores among any patients between enrollment and placebo week, thus precluding any withdrawal effects on the six subjects who had been treated with stimulants and the one subject treated with atomoxetine (Supplementary Table 4). This indicates that rebound effects, reported in stimulants[29], though not in atomoxetine[31], were adequately managed by the washout phase ranging from 5 days (stimulants) to 14 days (adrenergic drugs). A statistically significant decrease in mean CGI-S scores was observed during weeks 3–5 when compared to that at baseline. All subjects showed a decrease in symptom severity as measured by the CGI-S scores during dose escalation. The mean CGI-S score of all subjects at baseline was 4.83 (moderately to severely ill), decreasing to 3.86 (mildly to moderately ill) after receiving 400 mg BID of NFC-1 week 5. CGI-I scores also showed dose-dependent decrease in symptom severity during dose escalation from 50 mg BID to 400 mg BID.

At baseline most subjects were rated as moderately to severely impaired (mean CGI-S 4.83), indicating that their ADHD symptoms were disruptive enough to be causing functional impairment. After 1 week of fixed placebo followed by 4 weeks of dose escalation from 50 mg BID to 400 mg BID, over 80% of subjects showed clinically significant improvement in ADHD symptoms with mean CGI-S score of 3.86, mildly to moderately impaired. Significant improvement was also observed in CGI-I scores as compared to placebo therapy (Fig. 1 and Table 4). The effect was most marked in Tier-1/Tier-2 mGluR mutation positive subjects (*P* < 0.001), generated in a double-blinded assessment. Parental Vanderbilt scores also showed significant improvement for subjects with mGluR Tier 1 variants.

Improvement increased as dose was increased, suggesting dosages starting at 100 mg BID (3–5 mg/kg per day) may be required for clinical response. These results are likely to underestimate the response rate since maximum dosing of 400 mg BID was achieved in only 64, 71 and 66% of Tiers 1, 2 and 3, respectively. The level of comorbid symptoms could also affect the ability to rate the Vanderbilt scale during such a short therapy course. As the baseline Vanderbilt scores were slightly lower in Tier-3 subjects, than in Tier-1/Tier-2 subjects (Table 4), it is conceivable that the relatively lower response rate in Tier-3 subjects may be attributed to those differences. However, the CGI-I baseline scores were more similar and subjects with Tier-1 and Tier-2 mutations demonstrated more robust response in CGI-I than Tier-3 subjects (Figs. 1 and 2 and Table 4). Higher baseline ADHD-RS scores could be considered for inclusion in future studies.

The half-life (*T*$_{1/2}$) of NFC-1 ranged from 4.06 to 6.99 h with an average *T*$_{1/2}$ of 4.82 h across all four dose ranges. These results are similar to those previously reported in Japanese adults[32]. Other pharmacokinetic parameters (Table 2) were likewise comparable to those previously reported[32], in keeping with previous reports showing that NFC-1 is a stable compound that is excreted for the most part unchanged through the kidneys and that there is little, if any, enterohepatic circulation taking place. Measures of the two major drug metabolized of NFC-1 demonstrated levels and excretion profiles that were comparable to those previously reported[32]. Detectable levels of NFC-1 were observed in the subject's plasma at each visit, that allowed us to validate drug compliance which was in keeping with the log files of the drug intake.

We conclude that in adolescent patients with ADHD and mutations impacting the mGluR network genes, NFC-1 is safe and well-tolerated. Although the analysis of study efficacy is limited to early stage data, patients receiving NFC-1 showed significant clinical improvement in ADHD symptoms as clinically assessed by global rating scales. Response occurred in a dose-related manner, minimally requiring 100 mg BID for significant effect. Furthermore, as hypothesized, the degree of response observed was associated with the presence of specific gene disruptions in mGluR network genes as the most significant improvement in global rating scales were observed in patients in Tier 1/Tier 2. This study supports the continued investigation of NFC-1 in the treatment of ADHD, but also emphasizes the value of genetic prioritization and target therapy in ADHD treatment regimen selection. We recognize that alternative mechanisms explaining the therapeutic efficacy of Fasoracetam may exist. For example, Fasoracetam also affects the cholinergic pathways and thus may also prove to be effective in those ADHD subjects with

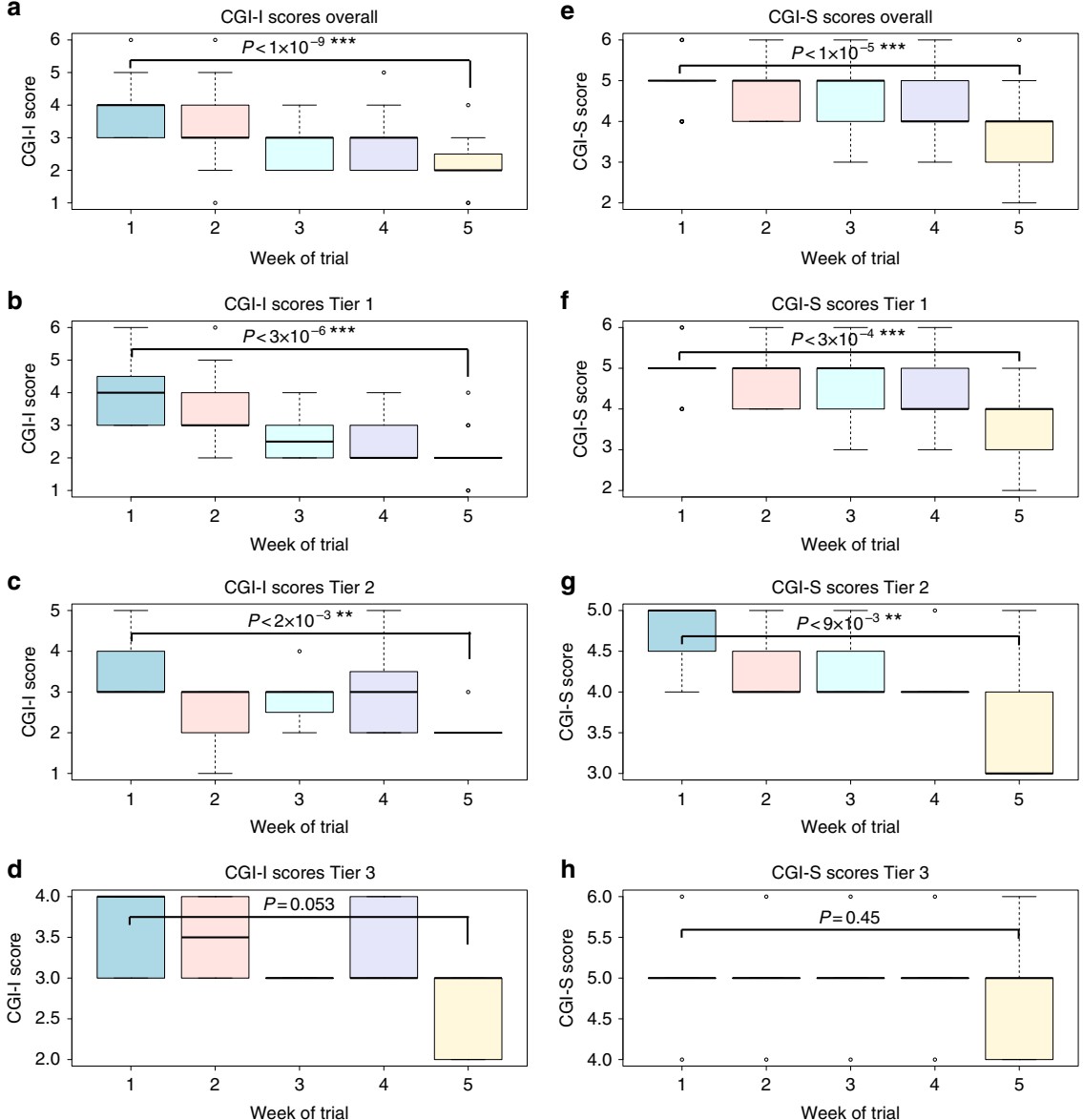

**Fig. 1** Week by week box-and-whisker plots showing the results for the CGI-I (**a–d**) and CGI-S (**e–h**) scaled inventories for all study subjects (**a**, **e**) or stratified by genetic tiers (Tier 1 (**b**, **f**), Tier 2 (**c**, **g**), and Tier 3 (**d**, **h**)). P-values denote results of the paired Student's t-test between results from study baseline (week 1) and final (week 5). $N = 30$. The edges of the box plots denote 25 and 75% tiles, while the solid black horizontal line denotes the cohort median. Upper and lower whiskers denote the limits of the nominal range of the data inferred from the upper and lower quartiles (Methods section) and plotted points are outliers from these ranges

variants in that pathway. It is also possible that activating NMDA receptors may compensate for dysfunctional cholinergic neuro-transmission[33]. Further studies are needed to examine the molecular and neurobiologic basis for our observations.

As this is a phase I clinical trial, the sample size is limited, recruitment was from a single tertiary care site, and treatment duration is limited. However, the study design consisted of a single-blinded placebo group of 1 week duration and the study was double-blinded as to mutation tier status which was unblinded at the end of study. With the exception of patient and family blinding as to which week patients received placebo, patients were otherwise aware of the medication given and that there was dose escalation taking place during the study. Nevertheless, given the safety of NFC-1 and the promising results from this cohort, this study supports the continued investigation of NFC-1 in the treatment of ADHD subjects with mGluR risk variants.

## Methods

**Study outline.** This is a single-site, single-blind, fixed placebo-controlled Phase 1 clinical trial to evaluate the safety, tolerability, plasma concentration profile and targeted efficacy of orally administered NFC-1 in adolescent (12–17 years) ADHD patients exhibiting mutations in genes within the GRM network. Primary objectives were to evaluate the safety and tolerability of orally administered NFC-1 and characterize its pharmacokinetic parameters. A secondary objective was to explore the dose-response relationship of NFC-1 on ADHD severity and global measures and determine effect size of specific GRM network genes on ADHD based on responsiveness of patients to NFC-1.

**Participants.** From over 200 patients screened for mGluR mutations using a SNP array, 30 patients with ADHD harboring disruptive copy number variants (CNVs) within or nearby one or more mGluR network genes (Table 5) were recruited from the Center of Applied Genomics (CAG) at The Children's Hospital of Philadelphia (CHOP). The study participants included ADHD cases who had previously participated in a large-scale genomics study at CAG/CHOP ("Genetics of Complex Pediatric Disorders") and had authorized recontact for future studies. The study was conducted at the Jefferson University (TJU) Clinical Research Unit for the PK

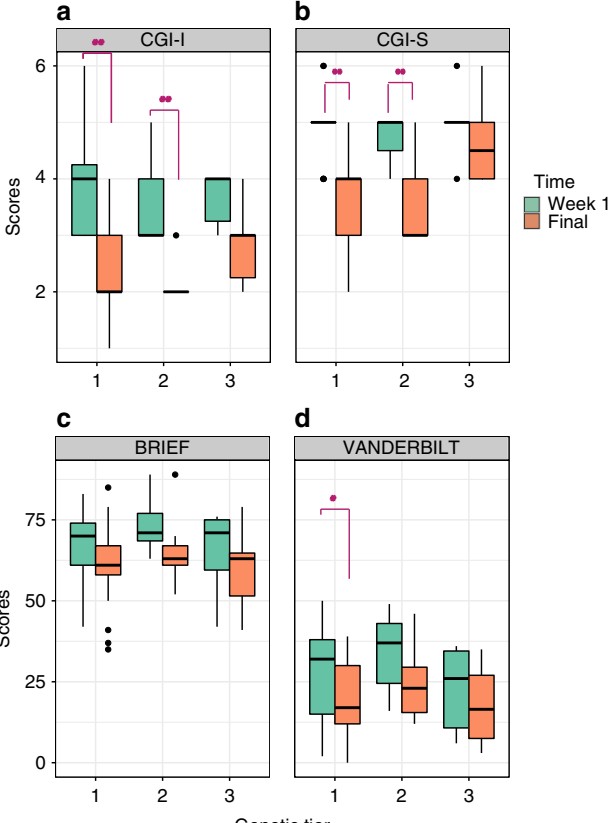

**Fig. 2** Net differences in the distribution of a CGI-I, b CGI-S, c Vanderbilt, and d Brief global scales between week 1 (placebo; green) and the final (max dose; orange) week of the study shown as box plots. Patients were stratified by genetic tier group allocation. *P*-values denote results of the paired Student's *t*-test between results from study baseline (week 1) and final (week 5) for each tier. Statistically significant comparisons are highlighted with red asterisks. N = 30. The edges of the box plots denote 25 and 75% tiles, while the solid black horizontal line denotes the cohort median. Upper and lower whiskers denote the limits of the nominal range of the data inferred from the upper and lower quartiles (see Methods) and plotted points are outliers from these ranges

part of the study and TJU Department of Psychiatry for the dose escalation part of the study. The study was approved by the Institutional Review Board of Thomas Jefferson University. Informed consents were obtained from all parents and assents from all study subjects. The study IND was approved on 7 November 2014; institutional review board approval was obtained on 11 December 2014; FPI was obtained on 23 January 2015, and LPO was obtained on 17 May 2015.

**Inclusionary criteria.** Male and female patients, ages 12–17 years, of any ancestry, diagnosed with ADHD as defined by DSM-5 and a Vanderbilt ADHD rating scale score ≥ 16, genotyped to have disruptive CNVs in the mGluR gene networks were eligible to participate (Supplementary Materials; Supplementary Figs. 1a and 1b).

**Exclusionary criteria.** Any clinically significant illness, which, in the opinion of the investigator, may confound the results of the study, pose additional risk to the patient by their participation, or prevent or impede the patient from completing the study, pregnancy, positive urine for illicit drugs (including marijuana), or history of drug or alcohol abuse within the last 3 years.

**Study drug.** NFC-1 (fasoracetam monohydrate) was administered per os as a single dose during the pharmacokinetic study and twice-daily (BID) during the dose-escalation study. NFC-1 was delivered in size 1, dark blue capsules containing 50 mg NFC-1 or size 1, dark blue capsules containing 200 mg NFC-1. Matching placebo capsules containing microcellulose instead of NFC-1 were provided during the week of placebo administration. The capsules were stored at controlled room temperature of 20–25 °C (68–77°F). Twelve-month stability data at room temperature showed no evidence of decay of the active substance.

**Dosing.** All study subjects completed a 24-h PK study. This was followed by an open-label, 5-week dose-escalation study, starting with a fixed single-blind placebo week (week 1). All ADHD medications were discontinued with a washout phase ranging from 5 days (stimulants) to 14 days (adrenergic drugs). The 24-h PK study included five groups with six subjects/group, where each group was given the following respective single doses: 50, 100, 200, 400 and 800 mg. Three patients were initially enrolled for PK studies and shown to tolerate 50 mg of the drug well prior to enrolling the next three subjects to complete dose group 1 of 50 mg. The next six subjects completed 100 mg and tolerated the study drug well. The DSMB approved the 3rd study group of 6 subjects to advance to 200 mg and the 4th set of six subjects to 400 mg, with the last group of 6 subjects approved by the DSMB to receive 800 mg (Supplementary Figs. 2 and 3). All study subjects tolerated their study medications well during the PK part of the study. The patients subsequently received a placebo BID (week 1) followed by NFC-1 50 mg BID in week 2, 100 mg BID in week 3, 200 mg BID in week 4, and 400 mg BID in week 5. Safety data was reviewed by the study PI, study team and the DSMB after each dose cohort before proceeding with each dose increase. Patients were blinded as to which week (out of the 5 total weeks) they received placebo.

**Study measures.** The PK measures included 11 parameters of active drug and two major metabolites at each time point during the initial pharmacokinetic portion of the trial. Blood samples were collected just prior to dosing and at 0.5, 1, 1.5, 2, 3, 4, 6, 8, 12, and 24 h after administration of the first dose of NFC-1. The concentrations of NFC-1 and two metabolites (LAM-79 and LAM-163) in plasma were determined by a validated ultra-performance liquid chromatography - tandem mass spectrometry assay with $D_{10}$-NFC-1 as an internal standard. The following precursor to product ion transitions were used for quantitation: $m/z$ 197.2 → 84.2 (NFC-1), $m/z$ 213.2 → 83.9 (LAM-79), $m/z$ 213.2 → 84.0 (LAM-163), and $m/z$ 207.2 → 84.2 ($D_{10}$-NFC-1). The analytical method was selective and linear over the concentration range of 10 to 2,500 ng/mL for all three compounds in plasma. Precision of the NFC-1 assay, as determined by percent coefficient of variation, was 5.13–7.16 % (within-day) and 3.81–9.99% (between-day), and accuracy was within 93.5–107% based on quality control samples. Precision of the LAM-79 assay was 5.81–9.11% (within-day) and 4.61–7.72% (between-day), with accuracy within 103–109%. Precision of the LAM-163 assay was 4.29–7.90% (within-day) and 3.28–10.2% (between-day), and was accuracy within 100 to 112%. PK parameters, including $C_{max}$, $T_{max}$, and $AUC_{0-24h}$, were estimated by the methodology of Gibaldi and Perrier[34]. $C_{max}$ and $T_{max}$ were obtained by visual inspection of individual patient plasma profiles. Half-life was estimated by 0.693/$k_e$, where $k_e$ is the slope of the terminal elimination phase from linear regression of log-transformed concentration values on time. AUC(0–24) was estimated by the linear trapezoidal method and AUC(0–∞) by addition of area extrapolated to infinity via the regression described above. All calculations were performed with the R programming language[35].

CGI-I/CGI-S scores determined by the psychiatric evaluators, together with parental Vanderbilt ratings, were the key measurements used to assess drug efficacy of NFC-1. The efficacy measures performed during the dose escalation study are detailed in Supplementary Figs. 2 and 3. Parental Brief rating scales were also included as secondary measures of NFC-1 efficacy. Wechsler Abbreviated Scale of intelligence, together with overall assessment of the children's neurocognitive function and ADHD-comorbid condition severities, including anxiety, autism spectrum disorder and mood swings, were also recorded by the psychiatric evaluators. Actigraphy data was also collected from all subjects at each visit for the duration of the study (Table 6).

The safety measures and logging of all adverse events (spontaneous reports and rating scales; Pittsburgh Side Effect Scale[36] and the Columbia Suicide Scale[37] were evaluated throughout the study and graded in terms of severity (mild, moderate, or severe) and relationship to study treatment (unrelated, possibly related, or definitely related) recorded. Serum NFC-1 levels were measured during each week of the dose-escalation study to ensure adherence. At baseline and after each week of treatment, each subject had a physical exam, electrocardiogram, and safety laboratory test (complete blood count, clinical chemistry and urinalysis).

**Genotyping and genetic tier stratification.** After informed consents were obtained, blood samples were obtained and deoxyribonucleic acid (DNA) was isolated from eligible adolescents age 12–17 years and genotyped using the OMNI-2.5 M genotyping assay (Illumina, San Diego), performed in the CAP-certified genotyping laboratory at The Children's Hospital of Philadelphia. The OMNI-2.5 M oligonucleotide array effectively captures the target CNVs in the 79 Tier-1 mGluR genes of interest and has sufficient coverage across the additional 200 Tier-2 and 600 Tier-3 mGluR genes, as previously reported[5,6].

In brief, 250 ng of genomic DNA was used to genotype each sample according to the manufacturer's guidelines. This implies, amplification of the patient DNA and hybridization to a reference genome. Upon washing, the samples are scanned to capture over 2.5 M SNPs dispersed throughout the genome. Cluster plots are generated to separate the 3 genotype states, namely homozygous allele (AA), homozygous allele (BB) or heterozygous state (AB).

The CNV quality control measures were performed on the genotyping data based on statistical distributions to exclude poor quality DNA samples and false positive CNVs. The first threshold was the percentage of attempted SNPs that were successfully genotyped. Only samples with call rate > 98.5% will be included in the

**Table 5 mGluR gene network variants impacting enrolled patients**

| Patients | Chr:Start-Stop(hg19) | CNV del/dupl | StartSNP | EndSNP | GRM contributing |
|---|---|---|---|---|---|
| *Tier-1 genes* | | | | | |
| Subject 1 | chr7:126447564-126535600 | Del | Startsnp = rs4731330 | Endsnp = rs12669064 | GRM8 |
| Subject 2 | chr17:76573316-76595131 | Dupl | Startsnp = rs752811 | Endsnp = rs11651207 | TK1 |
| Subject 3 | chr17:43661362-43685925 | Dupl | Startsnp = kgp3353562 | Endsnp = rs28713506 | CRHR1 |
| Subject 4 | chr7:86940652-86945816 | Del | Startsnp = kgp11792873 | Endsnp = kgp13383594 | GRM3 |
| Subject 5 | chr19:49073874-49075277 | Dupl | Startsnp = kgp21488201 | Endsnp = rs2544796 | RUVBL2 |
| Subject 6 | chr7:154885237-154889184 | Del | Startsnp = rs1436818 | Endsnp = rs1730186 | DPP6 |
| Subject 7 | chr11:88308509-88588641 | Del | Startsnp = rs160520 | Endsnp = rs7938157 | GRM5 |
| Subject 8 | chr8:90836233-90858597 | Del | Startsnp = kgp6180974 | Endsnp = kgp20067788 | 18p Del Syndr/RIPK2 |
| Subject 9 | chr7:125555887-125556965 | Del | Startsnp = kgp4046066 | Endsnp = kgp3698082 | GRM8 |
| Subject 10 | chr17:44166604-44347946 | Dupl | Startsnp = kgp3045933 | Endsnp = kgp13853487 | CRHR1 |
| Subject 11 | chr3:1936873-1943773 | Del | Startsnp = rs12637547 | Endsnp = kgp5837095 | CNTN4 |
| Subject 12 | chr7:153296249-153297555 | Del | Startsnp = rs10250553 | Endsnp = kgp8411417 | DPP6 |
| Subject 13 | chr22:18874965-21464479 | Del | Startsnp = kgp15094602 | Endsnp = kgp7040282 | 22Qdel - RANBP1 |
| Subject 14 | chr17:43907896-43913030 | Dupl | Startsnp = rs16940665 | Endsnp = rs16940686 | CRHR1 |
| Subject 15 | chr7:125435694-125576985 | Del | Startsnp = kgp9704135 | Endsnp = rs11767202 | GRM8 |
| Subject 16 | chr14:96683973-96688669 | Del | Startsnp = rs945040 | Endsnp = rs4905469 | BDKRB2 |
| Subject 17 | chr22:19421229-20308800 | Dupl | Startsnp = kgp6235116 | Endsnp = rs1210829 | 22Qdupl - RANBP1 |
| *Tier-2 genes* | | | | | |
| Subject 18 | chr18:637631-717229 | Del | Startsnp = rs13381806 | Endsnp = rs7235957 | C18orf56/ENOSF1 |
| Subject 19 | chr17:44173505-44347946 | Dupl | Startsnp = kgp7830309 | Endsnp = kgp13853487 | MAPT |
| Subject 20 | chr9:139743497-139755375 | Dupl | Startsnp = rs12555238 | Endsnp = kgp1725313 | TRAF2 |
| Subject 21 | chr6:33496513-33525680 | Del | Startsnp = kgp11013882 | Endsnp = rs210170 | GRM4 |
| Subject 22 | chr19:43513659-43594955 | Del | Startsnp = rs11668932 | Endsnp = kgp6508883 | CIC |
| Subject 23 | chr5:140225908-140227999 | Dupl | Startsnp = rs7730895 | Endsnp = kgp22520363 | PCDHA1-A9 |
| Subject 24 | chr17:7264717-7265681 | Del | Startsnp = kgp3155353 | Endsnp = kgp3898700 | SHBG |
| *Tier-3 genes* | | | | | |
| Subject 25 | chr11:1857042-1864645 | Dupl | Startsnp = rs907605 | Endsnp = kgp253861 | SYT8/TNNI2 |
| Subject 26 | chr2:201319521-201465841 | Dupl | Startsnp = kgp1394853 | Endsnp = rs16833843 | AOX1/KCTD18/SGOL2 |
| Subject 27 | chr5:102123728-102141765 | Dupl | Startsnp = kgp6255090 | Endsnp = kgp8236099 | PAM |
| Subject 28 | chr5:102123728-102141765 | Dupl | Startsnp = kgp6255090 | Endsnp = kgp8236099 | PAM |
| Subject 29 | chr19:7251101-7252844 | Del | Startsnp = rs11671297 | Endsnp = kgp10769783 | INSR |
| Subject 30 | chr2:110856809-110887808 | Del | Startsnp = kgp14587515 | Endsnp = kgp1736195 | MALL |

**Table 6 Results from actigraphy analysis**

| Measure | 50 mg | 100 mg | 200 mg | 400 mg | Model *P*-value |
|---|---|---|---|---|---|
| Mean Hourly kcals | -1.93 | 0.85 | -2.08 | -3.84 | 0.0099 |
| Bouts per day (*n*) | 0.05 | 0.5 | -0.03 | -0.34 | 0.29 |
| Sedentary bouts (*n*) | 0.25 | 0.9 | 0.95 | 0.48 | 0.34 |
| Percent in MVPA | 0 | 0 | -0.01 | -0.02 | 0.0053 |
| Mean MVPA per hour | -0.11 | 0.04 | -0.55 | -0.91 | 0.0035 |
| Vector magnitude average counts | -3.95 | -4.95 | -24.31 | -36.63 | 0.0009 |

Actigraphy results for study patients by drug dose; moderate to vigorous physical activity (MVPA)

**CNV analysis.** The genome-wide intensity signal must have as little noise as possible. Only samples with the standard deviation (SD) of normalized intensity (Log R Ratio (LRR)) < 0.25 were included. Samples of different ethnicities based on hierarchical clustering of AIMs genotypes were separated from each other and called separately. Wave artifacts that roughly correlate with GC content that result from hybridization bias of low full length DNA quantity are known to interfere with accurate inference of copy number variations. Thus, only samples where the correlation of LRR to wave model ranges between $-0.2 < X < 0.4$ will be accepted.

The simultaneous analysis of intensity data and genotype data in the same experimental setting establishes a highly accurate definition for normal diploid states and any deviation thereof. In contrast with aCGH which relies on intensity data alone for CNV calls, the Illumina SNP array platforms provide both genotype and intensity data for each SNP marker. The PennCNV algorithm was used to generate CNV calls and data analysis were standardized as described previously[39]. (Three different CNV states were called, including homozygous deletions (copy number, or CN = 0), hemizygous deletions (CN = 1), duplications (CN = 3 or more). Supplementary Table 5 presents the CNV regions for the top 79 most significant gene loci of interest harboring CNVs (deletions or duplications) and includes the same information for Tier-2 and Tier-3 genes.

All CNVs identified were validated visually and/or experimentally based on the confidence of the CNV call to ensure validity of the results reported (experimental validation has been performed previously or the 79 CNV genes under study[5]. As shown previously, for the 79 mGluR/GRM network genes, accuracy of the CNV calls detected on the SNP array is over 95%. Subsequent Taqman validation raised the accuracy of the CNV calls identified to 100%.

**Statistical and network analysis.** Quantitative variables were summarized using descriptive statistics. Continuous variables were presented as N, mean and median, standard deviation and range. Categorical variables were presented using frequencies and percentage. Analyses of variance (ANOVA) were used to compare treatment response at the respective weeks from week 1 to week 5. Where not specified explicitly, a Student's *t*-test was used. Analysis was performed using SPSS and R[35,38].

Genetic stratification of mGluR genes is delineated in genetic tiers in keeping with previous mGluR CNV association results (tier-1: 79 genes; tier-2: 200 genes; tier-3: 600 genes; Supplementary Materials)[5,6].

For Box-and-Whisker Plots, upper and lower whiskers denote the limits of the nominal range of the data inferred from the upper and lower quartiles, representing the smaller of the maximum score and the value of (75% tile+1.5×interquartile range) and the larger of the minimum score and the value of (25% tile −1.5×interquartile range), respectively. Additional points are outliers beyond these ranges. Network analysis were performed in StringDB (https://string-db.org) using program defined parameters.

**Actigraphy.** *Actigraphy watches*: Each participant was given an Actilife watch. The data was downloaded from the watch each week and the watches were recharged while participants were given their weekly dose, and a clinical interview.

*Data analysis*: Data was loaded into the Actilife software for further analysis. Sixty second epochs were used as the unit for analysis. The default algorithm in the software was used to detect off-watch times, and these periods were removed from further evaluation.

The Actilife software calculates a number of variables that measure energy expenditure, bouts of activity, and the vigorous nature of the activity. Bouts of activity are measured by an algorithm developed by Freedson[4] that has been calibrated in children and adolescents.

*Variables examined included*: average number of kCals used per hour, number of bouts of activity per day (measured using the Freedson et al., algorithm), total time of bouts, number of sedentary bouts, p of time in moderate to vigorous physical activity (MVPA)[40].

For each variable, a linear mixed model using was fit using the lme4 package in R where each variable was treated as a fixed effect, and regressed against the fixed effect of dosage, while treating subject as a random effect. The covariates of age, sex and weight were also included as fixed effects. A model with all covariates was fitted, and then a model with dosage added as a fixed effect and modeled as a factor. ANOVA using maximum likelihood was used to test for a significant difference in fit between the two models. The analysis compares whether there are differences overall between week 1 and the subsequent weeks.

To test if there was an interaction between dosage of the drug and the type of CNV carried by the participant, the difference in fit between models that included dosage and genetic tier in addition to the covariates, to one that also fit an interaction term of dosage×genetic tier.

**Data availability**. The data sets generated during and/or analyzed during the current study are available from the corresponding author on reasonable request.

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

## Acknowledgements

We thank the patients and their families for their participation in this study. We thank Phillip R. Harper and Rita Harper for their funding support to NeuroFix for the study execution. We thank David H. Moskowitz and Marian Moskowitz for their voluntary help with contractual agreements. We thank Maria B. Ivarsdottir for her voluntary accounting work. We thank David Fitts, Liza Squires, Garry Neil and Aevi Genome Medicine, Inc. for providing the analysis of the PK data measures for the study and for their review of the manuscript. We thank Donna F. Stroup, PhD, MSc for her statistical support. The study was funded by neuroFix Therapeutics Inc., a spin-off company from the Children's Hospital of Philadelphia. YRL is supported by the Paul and Daisy Soros Fellowship and the NIH F30 NRSA Individual Fellowship.

## Author contributions

J.E., W.K.K., and H.H. conceived the project, designed the experiments and supervised the study. J.E., G.U., C.K., N.De.J.-R., L.L., R.C., T.W., C.K., K.T., B.S., S.H., B.K. J.P., J.D., J.M., E.S., G.M., A.Z., A.W. and W.K.K. performed the experiments. J.E., W.K.K., C.K.,

Y.R.L. and H.H. wrote the manuscript and all authors contributed to manuscript editing and approved the final version of the article. All authors agree with the results and conclusions of this article. Y.R.L., A.A., and E.B. contributed to reagents, materials or analysis tools and helped with data analysis.

## Additional information

**Competing interests:** H.H. is the founder of NeuroFix, served on its Advisory Board and has stock or equity in neuroFix Therapeutics LLC, which is now owned by Aevi Genome Medicine Inc. H.H. was not involved with the evaluation of the study participants. All of the clinical evaluations were performed by J.E. and W.K.K. and their staff, none of whom have any competing interests, or any affiliation with neuroFix. All of the statistical analysis was done based on a predesigned SAP by an independent statistician as work for hire from a locked database. The remaining authors declare no competing financial interests.

