## [Peer review file · Nature Communications]

Editorial Note: Parts of this Peer Review File have been redacted as indicated to maintain patient confidentiality.

Reviewers' comments:

Reviewer #1 (Remarks to the Author):

The authors follow-up their previous work to present a preliminary trial of an mGlu5R activator in adolescents with ADHD. A total of 30 were included in the analysis with 29 patients completing all timepoints in this ascending dose study. The primary objective was to evaluate safety and tolerability of NFC-1. The secondary objective was to explore the dose response relationship with ADHD severity and determine the effect size of specific GRM-network genes. These are important and valuable aims.

The treatment of ADHD symptoms has been completely dominated by catecholaminergic compounds (and I include modafinil here), but there is evidence of glutamatergic abnormalities including important work by the authors. The trial of NFC-1 was conducted as an open-label 5-week dose escalation study.

The background for the study highlights an MRS study in ADHD showing lower frontal glutamate (MacMaster et al. ref. 11). Since this publication there has been a meta-analysis including the MacMaster study and the conclusions did not support this study (Perlov et al. 2009 World Journal of Biol Psychiatry 10). Please check this analysis and update your introduction accordingly.

The design of the study was confusing. In the methods (page 7, line 234) it states that patients "received a placebo BID (week 1) followed by NFC-1..." This is repeated in the table legend (line 339). However, on line 237 it states that patients were blind as to which week they would receive placebo. I assume that this means the placebo week was fixed to week 1 and the patients did not know this? A little clarification would be helpful to avoid possible confusion. The delivery of placebo on week 1 (assuming this is correct) follows medication withdrawal of 5 days for stimulant users, up to 2 weeks for adrenergic drug users. Please add in a table or the text if patients were withdrawn from mono-therapy or if patients were on >1 medication. Also, I did not see anywhere in the main manuscript or supplementary materials a list of which patients were on which medication or medication class. This would be a really important addition. The concern here is that there is medication rebound (Carlson & Kelly 2003 JCAP 13, 137-14). In this study 21.5% of patients who had withdrawn from stimulant medication had symptoms lasting about a week. The authors must address whether rebound could exaggerate symptom levels on placebo giving an artificially high effect size for subsequent drug. While rebound is not universal, in such a small sample size as this this could be a real risk. What would clarify this is symptom scores during the withdrawal period, but I did not see this as part of the protocol. Rebound could theoretically affect the primary outcome measure too, although I do not know of data that could speak to this directly. I also do not know of data on rebound to adrenergic treatments. The fixed order design means that this must be addressed.

Reviewer #2 (Remarks to the Author):

This paper presents results of a Phase 1 pharmacokinetic and efficacy/safety study of the mGluR agonist medication NFC-1 in adolescents with ADHD and mGluR genetic variability. The approach is highly innovative and the results are interesting. There are issues in the methodology and modeling of data which detract somewhat from the enthusiasm. Nevertheless, the study is important and deserves serious consideration. Comments by section follow.

Abstract. The design is not described sufficiently to clearly convey what was done. Absent a bit more information, the reader is likely to assume this is a more traditional clinical trial than it is. In this regard, it may seem curious to readers familiar with the ADHD literature that the change in Vanderbilt scores is not provided here in describing response to the two medications. CGI-S is

almost always a secondary measure; other clinical trials in ADHD use change in symptom measures to indicate the level of improvement. Even if a CGI measure is to be used in this section, CGI-I would normally be the one used to describe the magnitude of improvement.

Introduction. This section is generally clearly written and sets up the paper well. I have two suggestions. 1) The Visser study that is cited as a reference for prevalence is not a prevalence study; no diagnostic evaluations were undertaken or made. The authors should use a more representative prevalence study and also modify their text slightly, as the prevalence in the most recent US epidemiologic studies is ~8%, with age- and sex-related variations noted. 2) I was initially confused when I first read the paper as to whether this was a study of children with ADHD, or a study of a subgroup of children with ADHD with dysfunction in mGluR genes. The second to last paragraph suggests the study was done in a subgroup of you with ADHD, but the information in the last paragraph indicates that the study was done in children with ADHD (which I assume means a broader group). It is clear from the information presented later on that all subjects in this study had mGluR genetic variability. This should be clarified here.

Results. My comments here apply to the methods and results, as these are presented together. 1) The authors had not indicated previously that this study began with a week of placebo lead-in. This point comes up almost by accident in the description of AEs. Shouldn't this have been stated more clearly earlier? 2) I don't think the description of the SAE cases is clear enough. I am not sure what is meant by an "acting out" episode, and what might distinguish that from the kind of irritability, mood lability or even suicidal or aggressive behavior that can lead to psychiatric hospitalization (which is usually deemed to be possibly medication related). So either more or different information is required here. Regarding case #2, dizziness and possible loss of consciousness is as likely to occur from hypotension or another cardiac event as a neurological event. So the absence of neurological findings afterwards is only somewhat clarifying. As for case #3, it is certainly true that CPK can rise following exercise. But I assume all these subjects exercise, so one wonders why this happen in only one child? 3) It is unusual to have 4 primary efficacy measures in one study. And if so, wouldn't one have to correct for multiple analyses? 4) It seems odd that the authors describe CGI-I in this section of the text to illustrate improvement, yet they highlight CGI-S in the abstract. I am not sure whether there is a good enough reason for choosing either of these over the symptom rating, but even if the authors choose to use this approach, they should probably be consistent. 5) It is not clear which analyses of response the p values in line 118 refer to, since there were several variables analyzed and it does not specify here which this is. 6) The Vanderbilt scores at baseline seem very low for the Tier 3 subjects. This is probably a reflection of the entry criteria; I can't recall having seen an ADHD study with an entry criterion on an ADHD symptom scale of 16. I wonder if the degree of improvement in Tier 3 would be higher if the investigators looked only at a subgroup of subjects in this Tier with higher ADHD-RS scores. This is potentially important, because there is a major conclusion about the nature of genetic variants that are likely to be responsive to this drug. 7) The lack of substantive improvement in Vanderbilt scores with treatment overall, and especially in Tiers 2 and 3, is curious – as medication treatments for ADHD almost always show large improvement in symptoms. In this regard, the improvement seen in the actigraph measure is a help. Still, this raises the issue of whether the subjects studied here had other conditions as primary disorders with ADHD comorbid - with the medication helping these more primary conditions while having a secondary effect on ADHD symptoms. The authors might want to comment on this. 8) Using escalating doses sequentially in small cohorts is ideal for testing safety of a new drug but not as great for efficacy. Because some of the adolescents were treated in the early lower dose cohorts, and there might have been less response at lower dose, it is possible that the aggregate effects of the drug were under-estimated here. This could have impacted the relatively modest findings on the Vanderbilt. The authors would be within their rights to raise this point in discussion, and it would seem to serve their purpose well. 9) One potential problem about the genetic stratification analyses in relation to the sequential dose escalating design is that we can't be sure that the different mGluR variants were equally present across the different dose cohorts. Isn't it theoretically possible that more of the Tier 3 subjects were treated with lower doses, and might have had a more robust response if treated with higher doses? It might therefore be helpful to examine the distribution of dose in the different genetic variant groups. 10) The only one of the 4 efficacy measures that can

be referred to as an ADHD behavior inventory is the Vanderbilt. In this regard, Table 3A is mistitled. 11) I can't make sense of Figure 2. If this is retained it should be clarified or re-drawn. But is it even necessary?

Discussion. 1) I don't think the authors should make the statement that they used NFC-1 to study ADHD, as they do here. They studied a small genetically derived subgroup of youth who met criteria for ADHD in the context of other pediatric developmental disorders. This does not invalidate the importance of what was done. But it means they did not study the larger group of youth with ADHD. And this has implications for the interpretation of the study findings. 2) The improvements found here were illustrated better in global rating scales, as indicated in line 158, than specific symptom scales. To describe all of the scales used as global scales is not correct and potentially misleading. 3) The problem of focusing on the CGI scales to the relative exclusion of the Vanderbilt remains a problem here, as it was earlier on. 4) I am not sure how the authors can make a prediction of the likely required dose for improvement from these data, but maybe I am missing something here. 6) The last line of the discussion section strikes me as a bit problematic; as I see it, the data here do not shed light on whether or not NFC-1 would potentially be useful in the ~85% of youth with ADHD who do not have mGluR genetic variability, since these youth were not included in the current study.

Reviewer #3 (Remarks to the Author):

I have carefully reviewed the statistical methods both in the text and in the supplement. I have the following concerns about the statistical methods in the manuscript.

1. The statistical method adopted in this manuscript was not clearly illustrated. A lot of results or statements were presented with ambiguity.
2. Table 2A, it was unclear how those parameters were obtained.
3. Page 3, lines 93-95, the statement "None of the AEs were associated..." was not supported by any of the presented results. Please present results to support this statement.
4. Table 3A caption, it was mentioned that "P-values are calculated using a two-sided student's T-test". However, it was unclear whether it was a two-sample t-test or a paired t-test. Please be specific.
5. Table 3B, last column was labeled as "model p-value". What does it mean? How was it obtained? What "model" was used?
6. Figure 1, the violin plots were very confusing. For example, in the upper panel, how was the little black bar coming from? Was it from all tiers? The $p < 1 \times 10^{-9}$ was computed using all tiers? Same comments to the lower panel plot. In comparison, Figure 2 was more clear. But no p-values were presented.

Reviewer #4 (Remarks to the Author):

There is an obvious demand for new treatment options in ADHD. Considering that core symptoms of ADHD include inattention, it is logical to consider racetam type of cognitive enhancers, such as fasoracetam (NFC1/NS-105) .

Here, the authors describe a 5 wk open label study of 12-17 age old ADHD patients using up to 400 mg fasoracetam BID.

Major comment

1. The authors cite ref. 12 stating that the compound has undergone extensive phase I-III trials. However, refs. 12-14 are studies on rats, also citing studies on cholinergic drugs. Please rephrase or update references.

2. The exact molecular target(s) of fasoracetam appears to be unknown. Indeed, many different targets/transmitters have been implicated. Please explain/provide more details/appropriate references.

How do the other receptors/transmitters fit into their hypothesis?

3. Slightly different CGI-I responses were reported in different genetic subgroups (group size 6-17) of patients, with the largest Group giving lowest p-values. Please state if there were any significant differences between Tier 1-3 groups and how this was tested.

4. To understand the clinical results, the scores should be discussed in context. Are the effects in this open study larger than anticipated in a pure placebo study of similar design, taking into account ascertainment/assessor bias?

How strong is the effect, compared to a study on stimulants of similar design?

Are the modest effect of fasoracetam on core symptoms of ADHD consistent with earlier studies that found no nootropic effects of levetiracetam in autistic children (J Dev Behav Pediatr. 2002 Aug;23(4):225-30.)?

5. The authors propose that mGluRs play an important role in ADHD (affecting 30 out of 200 patients) and cite their previous CNV study implicating a glutamate network in ADHD. However, CNV findings in ADHD have been inconsistent, with larger samples giving less significant findings than the early studies on small samples. (Mol Psychiatry. 2016 Sep; 21(9): 1202). Moreover, in much larger GWAS analyses, SNPs in mGluR loci are not among the top hits (WCPG 2016). Together, this implies that alterations in mGluRs may be of minor importance in ADHD.

Please explain.

6. All humans have thousands of common and rare genetic variants that could contribute to ADHD symptoms. Classification of ADHD cases based on any CNV in mGluR networks appears simplistic.

7. Are there any functional studies indicating that the observed CNVs actually have an effect on glutamate signaling? In the absence of such data, the classification of all changes as being "disruptive" may be an exaggeration. Please consider rephrasing this.

8. The lack of effect of fasoracetam on core symptoms of ADHD appears to be consistent with earlier studies that found no nootropic effects of levetiracetam in autistic children: an open-label study J Dev Behav Pediatr. 2002 Aug;23(4):225-30.

Manuscript: Effects of NFC-1 in ADHD Adolescents Harboring Glutamatergic Gene Network Variants Disrupting mGluR Neurotransmitter Signaling”.

Reviewers' comments:

Reviewer #1 (Remarks to the Author):

The authors follow-up their previous work to present a preliminary trial of an mGlu5R activator in adolescents with ADHD. A total of 30 were included in the analysis with 29 patients completing all timepoints in this ascending dose study. The primary objective was to evaluate safety and tolerability of NFC-1. The secondary objective was to explore the dose response relationship with ADHD severity and determine the effect size of specific GRM-network genes. These are important and valuable aims.

The treatment of ADHD symptoms has been completely dominated by catecholaminergic compounds (and I include modafinil here), but there is evidence of glutamatergic abnormalities including important work by the authors. The trial of NFC-1 was conducted as an open-label 5-week dose escalation study.

1. The background for the study highlights an MRS study in ADHD showing lower frontal glutamate (MacMaster et al. ref. 11). Since this publication there has been a meta-analysis including the MacMaster study and the conclusions did not support this study (Perlov et al. 2009 World Journal of Biol Psychiatry 10). Please check this analysis and update your introduction accordingly.

We have updated the background by including the latest review (Naaijen et al 2015;52:74-88) that includes Perlov et al. 2009. Essentially, there is increased glutamatergic signaling in many, but not all the studies that include youths, and decreased glutamatergic signaling in adults. Glutamate changes in children treated with ADHD meds are also mixed with some studies showing decreases while the most recent one (Husarova et al 2014 – reviewed by Naaijen) shows increase with both methylphenidate and atomoxetine.

The design of the study was confusing. In the methods (page 7, line 234) it states that patients “received a placebo BID (week 1) followed by NFC-1...” This is repeated in the table legend (line 339). However, on line 237 it states that patients were blind as to which week they would receive placebo. I assume that this means the placebo week was fixed to week 1 and the patients did not know this? A little clarification would be helpful to avoid possible confusion.

Placebo week was fixed to week 1 and patients did not know which week they would be receiving the placebo. Changes to both the text and table legend have been made to clarify this.

The delivery of placebo on week 1 (assuming this is correct) follows medication withdrawal of 5 days for stimulant users, up to 2 weeks for adrenergic drug users. Please add in a table or the text if patients were withdrawn from mono-therapy or if patients were on >1 medication. Also, I did not see anywhere in the main manuscript of supplementary materials a list of which patients were on which medication or medication class. This would be a really important addition.

We thank the reviewer for addressing this and all medications have been included in Supplementary Table 4, entitled: Medications Discontinued at Enrollment and those maintained during the study.

Study participant	Stimulants	Atomoxetine	Guanfacine	Other Meds	During Study
101				Fluoxetine	
102				Clonidine-sleep	Clonidine
104	Vyvanse				
105	Adderall				
107	Vyvanse				
108				Lamictal	Lamictal
109		ATX			
113				Sertraline	Sertraline
114	Concerta				
115				Clonidine (sleep)	
118	Concerta				
119	Concerta				
121				Clonidine (sleep)	
125				Risperidone	

The concern here is that there is medication rebound (Carlson & Kelly 2003 JCAP 13, 137-14). In this study 21.5% of patients who had withdrawn from stimulant medication had symptoms lasting about a week.

We did not include the Carlson et al 2003 study since the study subjects were hospitalized youths, raters were not blinded, the definition of rebound was not clear and the study drug was short acting methylphenidate. We also note that a 24 hr. PK study followed 5 days of discontinuation of stimulants (and 2 wk discontinuation of adrenergic drugs) followed by 1 week of placebo before the lowest dose of active drug (50mg bid) was begun.

The authors must address whether rebound could exaggerate symptom levels on placebo giving an artificially high effect size for subsequent drug. While rebound is not universal, in such a small sample size as this this could be a real risk. What would clarify this is symptom scores during the withdrawal period, but I did not see this as part of the protocol. Rebound could theoretically affect the primary outcome measure too, although I do not know of data that could speak to this directly.

Medication rebound is a very important point. Supplemental Table 1 Study Flow-sheet shows that CGI-S scores were obtained at the time of enrollment (before wash-out phase). Text is revised to indicate that these scores did not change during placebo

week for any of the subjects including the 6 subjects who had been treated with stimulants and 1 with atomoxetine (Paragraph 2).

I also do not know of data on rebound to adrenergic treatments. The fixed order design means that this must be addressed.

We found no data on rebound and alpha agonists.

The atomoxetine studies did not detect any rebound.

Rebound would pose the greatest risk on driving a motor vehicle. Thus, the most rigorous study investigating rebound for long-acting stimulants included data on driving simulator as well as on-road driving measures (Cox et al. JCAP 2008: (18)1: 1-10). Driving did not worsen with long acting methylphenidate. In some males, worse driving was noted at 20 hours after AMPH ER was administered. There is no data beyond this time-point.

Reviewer #2 (Remarks to the Author):

This paper presents results of a Phase 1 pharmacokinetic and efficacy/safety study of the mGluR agonist medication NFC-1 in adolescents with ADHD and mGluR genetic variability. The approach is highly innovative and the results are interesting. There are issues in the methodology and modeling of data which detract somewhat from the enthusiasm. Nevertheless, the study is important and deserves serious consideration. Comments by section follow.

Abstract. The design is not described sufficiently to clearly convey what was done. Absent a bit more information, the reader is likely to assume this is a more traditional clinical trial than it is. In this regard, it may seem curious to readers familiar with the ADHD literature that the change in Vanderbilt scores is not provided here in describing response to the two medications. CGI-S is almost always a secondary measure; other clinical trials in ADHD use change in symptom measures to indicate the level of improvement. Even if a CGI measure is to be used in this section, CGI-I would normally be the one used to describe the magnitude of improvement.

We thank the reviewer for addressing this and the abstract has been expanded to include additional Vanderbilt and CGI data.

Introduction. This section is generally clearly written and sets up the paper well. I have two suggestions.

1) The Visser study that is cited as a reference for prevalence is not a prevalence study; no diagnostic evaluations were undertaken or made. The authors should use a more representative prevalence study and also modify their text slightly, as the prevalence in the most recent US epidemiologic studies is ~8%, with age- and sex-related variations noted.

Visser reference has been replaced with Polanczyk et al In. J Epidemiol 2014;43:434-442) and text has been changed accordingly.

2) I was initially confused when I first read the paper as to whether this was a study of children with ADHD, or a study of a subgroup of children with ADHD with dysfunction in mGLUR genes. The second to last paragraph suggests the study was done in a subgroup of you with

ADHD, but the information in the last paragraph indicates that the study was done in children with ADHD (which I assume means a broader group). It is clear from the information presented later on that all subjects in this study had mGluR genetic variability.

This should be clarified here.

The last sentence has been revised to reflect the continued investigation of NFC-1 in ADHD subjects with mGluR risk variants.

Results. My comments here apply to the methods and results, as these are presented together. 1) The authors had not indicated previously that this study began with a week of placebo lead-in. This point comes up almost by accident in the description of AEs. Shouldn't this have been stated more clearly earlier?

This has been clarified in the revised manuscript beginning with the abstract (see also response to comment from Rev-1).

2) I don't think the description of the SAE cases is clear enough. I am not sure what is meant by an "acting out" episode, and what might distinguish that from the kind of irritability, mood lability or even suicidal or aggressive behavior that can lead to psychiatric hospitalization (which is usually deemed to be possibly medication related). So either more or different information is required here.

[Redacted]

Regarding case #2, dizziness and possible loss of consciousness is as likely to occur from hypotension or another cardiac event as a neurological event. So the absence of neurological findings afterwards is only somewhat clarifying.

[Redacted]

As for case #3, it is certainly true that CPK can rise following exercise. But I assume all these subjects exercise, so one wonders why this happen in only one child?

Additional detail are included in text. [Redacted]

3) It is unusual to have 4 primary efficacy measures in one study. And if so, wouldn't one have to correct for multiple analyses? 4) It seems odd that the authors describe CGI-I in this section of the text to illustrate improvement, yet they highlight CGI-S in the abstract. I am not sure whether there is a good enough reason for choosing either of these over the symptom rating, but even if the authors choose to use this approach, they should probably be consistent.

As this was an exploratory study with respect to efficacy measures, we elected to use more than one primary efficacy measure. As these independent measures all demonstrated improvement the confidence level that the drug is having effect is higher independent of any multiple testing measures, which is what the study investigators had hoped observing. Both CGI-I and CGI-S are now highlighted in the abstract of the revised manuscript.

5) It is not clear which analyses of response the p values in line 118 refer to, since there were several variables analyzed and it does not specify here which this is.

Text is revised to clarify that this refers to CGI-I

6) The Vanderbilt scores at baseline seem very low for the Tier 3 subjects. This is probably a reflection of the entry criteria; I can't recall having seen an ADHD study with an entry criterion on an ADHD symptom scale of 6. I wonder if the degree of improvement in Tier 3 would be higher if the investigators looked only at a subgroup of subjects in this Tier with higher ADHD-RS scores. This is potentially important, because there is a major conclusion about the nature of genetic variants that are likely to be responsive to this drug.

The reviewer is correct that the baseline Vanderbilt scores are lower in Tier-3 subjects – see table 3A, pasted below for convenience. The baseline Vanderbilt score was 32, 37 and 26, respectively in Tier-1, Tier-2 and Tier-3 subjects (note, the Tier-3 score is 26, not 6).

	All	Tier 1	Tier 2	Tier 3
CGI-I	1.2E-09	3.1E-06	2.1E-03	5.3E-02
baseline	3.79(4) ±0.81	3.93(4) ±0.92	3.57(3) ±0.78	3.66(4) ±0.51
final	2.33(2) ±0.71	2.23(2) ±0.75	2.14(2) ±0.37	2.83(3) ±0.75
CGI-S	1.7E-05	3.3E-04	8.5E-03	4.5E-01
baseline	4.86(5) ±0.57	4.88(5) ±0.60	4.71(5) ±0.48	5(5) ±0.63
final	3.93(4) ±0.90	3.82(4) ±0.88	3.57(3) ±0.78	4.66(4.5) ±0.81
Vanderbilt (P)	1.0E-02	3.5E-02	1.8E-01	5.2E-01
baseline	28.7(32.5) ±13.9	28.7(32) ±14.5	33.8(37) ±12.4	22.8(26) ±13.7
final	19.7(17.5) ±12.2	18.5(17) ±12.2	24.4(23) ±12.3	17.6(16.5) ±12.8

BRIEF (P)	4.9E-02	1.8E-01	1.9E-01	4.7E-01
baseline	68.4(70.3) ±11.2	67.2(70) ±11.2	73.8(71.0) ±8.85	65.9(71.4) ±13.5
final	62.1(63.1) ±13.0	61.3(61.9) ±13.7	66.0(63.1) ±11.7	60.0(63.0) ±13.5

As the reviewer pointed out the lower Vanderbilt score in Tier-3 (i.e., 26, not 6) could result in some interference with the potential to improve on the Vanderbilt score in those subjects who started off with a lower baseline score, although they improved on average by 9.5 points (26 to 16.5). However, the CGI-I baseline scores were more similar and subjects with Tier-1 and Tier-2 mutations demonstrated more robust response in CGI-I than Tier-3 subjects. This has been clarified in the revised manuscript.

7) The lack of substantive improvement in Vanderbilt scores with treatment overall, and especially in Tiers 2 and 3, is curious – as medication treatments for ADHD almost always show large improvement in symptoms. In this regard, the improvement seen in the actigraph measure is a help. Still, this raises the issue of whether the subjects studied here had other conditions as primary disorders with ADHD comorbid - with the medication helping these more primary conditions while having a secondary effect on ADHD symptoms. The authors might want to comment on this.

Large effect size for ADHD treatment is primarily seen with the stimulants. Non-stimulants effect size is typically more limited.

Maximal dosing (400mg bid) showed the greatest effects on CGI-I and CGI-S, this does was maintained only for one week (#5) and 400mg bid was reached only in 64% in Tier 1, 71% in Tier 2 and 66% in Tier 3 (this was as a result of PK data not being available for the highest dose in the beginning so we couldn't advance the dose to top maximum in the first several kids). This likely contributed to the modest changes in parental Vanderbilt scores. The reviewer is correct that comorbid symptoms may also have played a role in response to NFC-1; we thank the reviewer for pointing this out and we have clarified this in the revised manuscript.

8) Using escalating doses sequentially in small cohorts is ideal for testing safety of a new drug but not as great for efficacy. Because some of the adolescents were treated in the early lower dose cohorts, and there might have been less response at lower dose, it is possible that the aggregate effects of the drug were under-estimated here. This could have impacted the relatively modest findings on the Vanderbilt. The authors would be within their rights to raise this point in discussion, and it would seem to serve their purpose well.

We appreciate this important observations and have included this in the text.

9) One potential problem about the genetic stratification analyses in relation to the sequential dose escalating design is that we can't be sure that the different mGluR variants were equally present across the different dose cohorts. Isn't it theoretically possible that more of the Tier 3 subjects were treated with lower doses, and might have had a more robust response if treated with higher doses? It might therefore be helpful to examine the distribution of dose in the different genetic variant groups.

We thank the reviewer for bringing this up. Tier 1 was actually the most disadvantaged with regarding to dosing. The % of subjects that reached maximum dosing in Tier 1, 2 and 3 were 64%, 71%, 66% respectively.

10) The only one of the 4 efficacy measures that can be referred to as an ADHD behavior inventory is the Vanderbilt. In this regard, Table 3A is mistitled.

The table title is revised to say “Study Measures”

11) I can't make sense of Figure 2. If this is retained it should be clarified or re-drawn. But is it even necessary?

We thank the reviewer for addressing this and we have revised the drawing of figure 2 to make the message more clear in the revised manuscript.

Discussion.

1) I don't think the authors should make the statement that they used NFC-1 to study ADHD, as they do here. They studied a small genetically derived subgroup of youth who met criteria for ADHD in the context of other pediatric developmental disorders. This does not invalidate the importance of what was done. But it means they did not study the larger group of youth with ADHD. And this has implications for the interpretation of the study findings.

In the first sentence of the discussion we state that the objective of the study is to study NFC-1 in adolescents with ADHD and disruptive mutations in genes within the mGluR network.

We have revised the 2nd sentence “Our study is the first to use NFC-1 to treat adolescents with ADHD who harbor mGluR risk variants”.

2) The improvements found here were illustrated better in global rating scales, as indicated in line 158, than specific symptom scales. To describe all of the scales used as global scales is not correct and potentially misleading.

This has been addressed in the revised manuscript

3) The problem of focusing on the CGI scales to the relative exclusion of the Vanderbilt remains a problem here, as it was earlier on.

Results from CGI-I, CGI-S and Vanderbilt are presented

4) I am not sure how the authors can make a prediction of the likely required dose for improvement from these data, but maybe I am missing something here.

CGI-I improvement was noted with increasing dosing with most significant improvement noted at the highest dose of 400mg bid, suggesting that the starting dosage of 50mg bid (the starting dose for this study to determine safety) may be too low for treatment (see figure and table below)

CGI-I: Results for the Clinical Global Impression – Improvement Scale are presented in figure below. These data summarize the proportion of Responders (CGI-I score of 1 or 2) among the total study population.

Figure CGI-I: Proportion of Responders at each Visit by Genetic Tier and Overall

Table: Clinical Global Impression – Improvement Scores: Proportions of Patients Responding at each study visit.

Tier	Week 1		Week 2		Week 3		Week 4		Week 5	
	N	%	N	%	N	%	N	%	N	%
1	16	0	17	24	15	53	16	56	16	81
2	7	0	7	43	7	29	6	50	7	86
3	6	0	6	0	6	0	6	0	5	40
Overall	29	0	30	29	28	45	28	55	28	83

6) The last line of the discussion section strikes me as a bit problematic; as I see it, the data here do not shed light on whether or not NFC-1 would potentially be useful in the ~85% of youth with ADHD who do not have mGluR genetic variability, since these youth were not included in the current study.

We have revised this to state “... Treatment of ADHD subjects with mGluR risk variants”

Reviewer #3 (Remarks to the Author):

I have carefully reviewed the statistical methods both in the text and in the supplement. I have the following concerns about the statistical methods in the manuscript.

1. The statistical method adopted in this manuscript was not clearly illustrated. A lot of results or statements were presented with ambiguity.

We must disagree with this statement from the statistical reviewer as all statistical results presented were based on pre-determined analyses proposed in the Statistical Analysis Plan (SAP) of the project prior to the conduct of the study, and were performed by an independent statistician (as work for hire) from a closed blinded database. We have included the SAP as an attachment to the paper for the purpose of giving the Reviewer an opportunity to review and confirm this (please see uploaded PDF entitled “SAP 05 27 2015_Final”).

2. Table 2A, it was unclear how those parameters were obtained.

We thank the reviewer for pointing this out. We have updated the methods section to include the following additional details:

PK parameters estimates were derived by the methodology of Gibaldi and Perrier. Cmax and Tmax were obtained by visual inspection of individual patient plasma profiles. Half-life was estimated by $0.693/k_e$, where k_e is the slope of the terminal elimination phase from linear regression of log-transformed concentration values on time. AUC(0-24) was estimated by the linear trapezoidal method and AUC(0-inf) by addition of area extrapolated to infinity via the regression described above. All calculations were performed with the R programming language.

3. Page 3, lines 93-95, the statement “None of the AEs were associated...” was not supported by any of the presented results. Please present results to support this statement.

A summary of all AEs per week (i.e., for placebo in week 1; for 50mg bid in week 2; for 100 mg bid in week 3; for 200 mg bid in week 4; and for 400 mg in week 5) are included in Supplemental Table 3 of the revised manuscript. As shown, the frequency of adverse events during the placebo week is comparable to those of any weeks on active drug. Neither the total number of events nor any individual event is more frequent in subjects on active drug at any given week of comparison.

4. Table 3A caption, it was mentioned that “P-values are calculated using a two-sided student’s T-test”. However, it was unclear whether it was a two-sample t-test or a paired t-test. Please be

specific.

Paired t-test was used and this has now been clarified in the revised manuscript, wherever mentioned.

5. Table 3B, last column was labeled as “model p-value”. What does it mean? How was it obtained? What “model” was used?

We refer the reviewer to the detailed supplemental methods included in the supplemental PDF, which includes a section describing how the model p-value was obtained (it is copied for easy reference by the reviewer). We have clarified that this refers to the ANOVA model p-value specifically in the table.

“For each variable, a linear mixed model using was fit using the lme4 package in R where each variable was treated as a fixed effect, and regressed against the fixed effect of dosage, while treating subject as a random effect. The covariates of age, sex and weight were also included as fixed effects. A model with all covariates was fitted, and then a model with dosage added as a fixed effect and modeled as a factor. Analysis of variance (ANOVA) using maximum likelihood was used to test for a significant difference in fit between the two models. The analysis compares whether there are differences overall between Week 1 and the subsequent weeks.”

6. Figure 1, the violin plots were very confusing. For example, in the upper panel, how was the little black bar coming from? Was it from all tiers? The $p < 1 \times 10^{-9}$ was computed using all tiers? Same comments to the lower panel plot. In comparison, Figure 2 was more clear. But no p-values were presented.

- 1) We apologize that the reviewer found interpreting Figure 1 to be difficult. We refer the reviewer to our figure legend for his/her question regarding the black bar: “Black dot denote the average across all groups at each time point with bars denoting the standard deviation (See text and methods).” Both are for all the tiers at a given time point.
- 2) As the reviewer felt that Figure 1 was too confusing, we have reduced the complexity of the plot. Now, in the revised Figure 1, both the top and bottom figure include three panels, one for each genetic tier. Within each panel are the CGI-I and CGI-S score distributions in the form of a classical boxplot.
- 3) The P-value in Figure 1 was obtained using a pairwise student’s t-test to compare the clinical inventory scores between weeks 1 and 5 for all subjects across all tiers.
- 4) We appreciate the reviewer pointing out that there were no p-values in Figure 2. We have ensured that the relevant p-values, notably those comparing week 1 versus week 5 using a paired student’s t-test are included in the revised figure. Patients are stratified by genetic tier.

Reviewer #4 (Remarks to the Author):

There is an obvious demand for new treatment options in ADHD. Considering that core symptoms of ADHD include inattention, it is logical to consider racetam type of cognitive enhancers, such as fasoracetam (NFC1/NS-105) .

Here, the authors describe a 5 wk open label study of 12-17 age old ADHD patients using up to 400 mg fasoracetam BID.

Major comment

1. The authors cite ref. 12 stating that the compound has undergone extensive phase I-III trials. However, refs. 12-14 are studies on rats, also citing studies on cholinergic drugs. Please rephrase or update references.

Ref. 12 has been moved to the preclinical studies.

2. The exact molecular target(s) of fasoracetam appears to be unknown. Indeed, many different targets/transmitters have been implicated. Please explain/provide more details/appropriate references.

Manuscript text has been revised to include more details on fasoracetam's MOA and the different targets involved.

How do the other receptors/transmitters fit into their hypothesis?

The various neurotransmitters don't function in isolation. The discussion in the text has been revised to include glutamatergic/cholinergic interactions.

3. Slightly different CGI-I responses were reported in different genetic subgroups (group size 6-17) of patients, with the largest Group giving lowest p-values. Please state if there were any significant differences between Tier 1-3 groups and how this was tested.

Comparisons between the 3 groups was not thought to be useful since the number of subjects in Tier-3 was small (n=6); Overall, there was greater improvement in Tier-1 and Tier-2 CGI-I scores than for Tier-3 CGI-I scores but the analyses is biased due to the unbalanced numbers between the Tiers (17, 7 and 6, respectively for Tiers 1, 2 and 3)

4. To understand the clinical results, the scores should be discussed in context. Are the effects in this open study larger than anticipated in a pure placebo study of similar design, taking into account ascertainment/assessor bias?

Medication effects are likely less than we expect given that not all subjects received optimal dosing.

How strong is the effect, compared to a study on stimulants of similar design?

Determining effect size for our study is not possible since not all subjects received optimal dosage.

We did not anticipate that NFC would have a similar effect size as stimulants (0.78-1.0) Leucht et al. 200(2)2012); however, in the subset of subjects who received full dose (400mg

bid), including 16 subjects, the effect size was 1.0, comparable to that of stimulants. We note that even a modest effect size is clinically relevant if a drug is well tolerated.

Are the modest effect of fasoracetam on core symptoms of ADHD consistent with earlier studies that found no nootropic effects of levetiracetam in autistic children (J Dev Behav Pediatr. 2002 Aug;23(4):225-30.)?

Similar to our study, in the Rugino & Samsoc study referenced above, levetiracetam was reported to decrease core ADHD symptoms of hyperactivity and impulsivity. Unlike our study, it did not decrease inattention.

The levetiracetam study included nootropic measures (language, verbal, perception) pre and post treatment. Our only assessment was an IQ test prior to starting the study so we can't comment on nootropic effect.

5. The authors propose that mGluRs play an important role in ADHD (affecting 30 out of 200 patients) and cite their previous CNV study implicating a glutamate network in ADHD. However, CNV findings in ADHD have been inconsistent, with larger samples giving less significant findings than the early studies on small samples. (Mol Psychiatry. 2016 Sep; 21(9): 1202).

In the Thaper et al. study referenced above, CNVs in ADHD samples were investigated for enrichment in SNVs implicated in schizophrenia and autism. They also investigated specific genes (ARC, NMDR and FMRP). The NMDR set included only a subset of gene variants that were investigated in the other studies.

Exome sequencing data was not available for the ADHD sample.

Moreover, in much larger GWAS analyses, SNPs in mGluR loci are not among the top hits (WCPG 2016). Together, this implies that alterations in mGluRs may be of minor importance in ADHD. Please explain.

Common SNPs that reside in the vicinity of mGluR genes are not able to tag these rare CNVs so we would not expect any of the common SNP variants on GWAS chips to show an association signal in relation with these rare CNVs, whether they are large or small CNVs. Thus, these are not going to show up as top hits in GWAS analyses. Our genome wide CNV analysis did, however, pick them up and demonstrated significant association with replication (Elia et al, Nat Genet, 2012).

6. All humans have thousands of common and rare genetic variants that could contribute to ADHD symptoms. Classification of ADHD cases based on any CNV in mGluR networks appears simplistic.

Classification of ADHD is complex and challenging.

Common variant conferring significant risk have yet to be identified. Those that confer minimal risk point to disruption in neuronal pathways. There continues to be increasing evidence that rare variants are the key players and while this may be simplistic, we believe that

grouping these based on the neuronal pathways is an important first step in targeting treatment.

7. Are there any functional studies indicating that the observed CNVs actually have an effect on glutamate signaling? In the absence of such data, the classification of all changes as being "disruptive" may be an exaggeration. Please consider rephrasing this.

The word "disruptive" has been removed in Paragraph 5 of discussion.

8. The lack of effect of fasoracetam on core symptoms of ADHD appears to be consistent with earlier studies that found no nootropic effects of levetiracetam in autistic children: an open-label study J Dev Behav Pediatr. 2002 Aug;23(4):225-30.

See #4 above.

REVIEWERS' COMMENTS:

Reviewer #1 (Remarks to the Author):

The authors have responded well to all my comments.

Reviewer #2 (Remarks to the Author):

The authors have been quite responsive to the reviews and the revised paper is much improved. I am left with one substantive comment, which probably cannot be resolved and is probably best handled as a limitation in the discussion. The rest of the comments are minor.

Major comment.

The relatively modest improvement in Vanderbilt scores overall, and in Tier 3 participants in particular, is lower than for both stimulant and non-stimulant medications for ADHD. If this finding were to stay as is in a larger study it would possibly be problematic. For a Phase I trial the findings here are certainly suggestive and support moving forward as the authors suggest. The authors' statement in the rebuttal letter that one only sees large effect sizes for stimulants is partially true but also a bit misleading. Effect sizes for approved non-stimulant medications for ADHD are in the range of 0.6-0.7 in children, not large but very solid. Also note that the original critique referred to large effects, not effect sizes.

Minor comments.

I believe the anchor point for 3 on the CGI-I is minimally improved, not moderately improved; there is no anchor point for moderately improved. There is a moderately ill anchor point on the CGI-S. I think the authors must have these confused (Line 117)

Improvement in ADHD symptoms would not be measured by CGI-S, but rather the symptom scale. One could say symptom severity here and be correct (line 181).

I am confused about the authors' statement regarding baseline CGI-I scores across the groups being comparable. CGI-I is a measure of improvement, and would not be relevant here. CGI-S would be a more appropriate measure to compare baseline severity across groups. (line 190)

In line 201 I would say that the CGI ratings reflected improved in symptom severity rather than symptoms, since symptoms are measured using a symptom scale such as the Vanderbilt.

Reviewer #3 (Remarks to the Author):

The authors have sufficiently addressed my comments.

Reviewer #4 (Remarks to the Author):

Reviewer #4, additional comments:

The Authors have responded to all of my comments.

Q1-3 and Q4: As suggested, the authors have rephrased some sentences and corrected a few errors in the original text.

Still, some remaining inaccuracies should be amended.

Without any experimental support, the word «disruptive» to classify the mutations is misleading and should be removed from the abstract and introduction.

Q4: The response to Q4 is still incomplete:

Could the clinical response simply be explained by a placebo effect or ascertainment bias in this

open label study?

Please explain this more explicitly.

Q5: As the genetic evidence supporting an altered glutamate signaling in ADHD is weak and controversial, the authors should be more cautious in their discussion of this mechanism. They correctly state that GWAS is not capturing rare events. However, if this signaling system is altered in such a large fraction of ADHD cases, it would not be unexpected to detect signals also in GWAS.

Although possibly captured by a network analysis, most of the genes listed in Table 1A are not GRMs. Many are encoding enzymes involved in diverse biological pathways without any obvious relationship with glutamatergic signaling. Consider relabeling this table.

Likewise, the term "mutations in genes within the mGluR network" should be reserved for genes directly related to glutamate signaling.

Manuscript: Fasoracetam in Adolescents with ADHD and Glutamatergic Gene Network Variants Disrupting mGluR Neurotransmitter Signaling

REVIEWERS' COMMENTS:

We thank all the reviewers for their thoughtful comments and suggestions that have helped us in further clarifying and improving the manuscript.

Reviewer #1 (Remarks to the Author):

The authors have responded well to all my comments.

Reviewer #2 (Remarks to the Author):

The authors have been quite responsive to the reviews and the revised paper is much improved.

I am left with one substantive comment, which probably cannot be resolved and is probably best handled as a limitation in the discussion. The rest of the comments are minor.

Major comment.

The relatively modest improvement in Vanderbilt scores overall, and in Tier 3 participants in particular, is lower than for both stimulant and non-stimulant medications for ADHD. If this finding were to stay as is in a larger study it would possibly be problematic. For a Phase I trial the findings here are certainly suggestive and support moving forward as the authors suggest. The authors' statement in the rebuttal letter that one only sees large effect sizes for stimulants is partially true but also a bit misleading. Effect sizes for approved non-stimulant medications for ADHD are in the range of 0.6-0.7 in children, not large but very solid. Also note that the original critique referred to large effects, not effect sizes.

The reviewer makes an important point regarding the modest improvement on Vanderbilt scores overall and in particular Tier 3 that could be potentially problematic in a larger trial. As Tier-3 mutations are further removed from the mGluR action/signaling, we do not plan to enroll individuals with Tier-3 mutations in future studies and rather focus on developing the drug in subjects with Tier-1 and Tier-2 mutations with the option of exploring efficacy in subjects with Tier-3 mutations and subjects without mutations in phase 4 studies if we succeed with development in Tier-1 and Tier-2 subjects. To complement the reviewers point, we have included the following sentence in the discussion section of the manuscript.

“Higher baseline ADHD-RS scores will be considered for inclusion in future studies”.

Minor comments.

I believe the anchor point for 3 on the CGI-I is minimally improved, not moderately improved; there is no anchor point for moderately improved. There is a moderately ill anchor point on the CGI-S. I think the authors must have these confused (Line 117)

The reviewer is correct regarding anchor points for CGI:

1 very much improved

2 much improved

3 minimally improved

4 no change

5 minimally worse

6 much worse

7 very much worse

Our baseline score was 3.79 and we indicated this as ‘minimally improved to no change’

Our score at week 5 after treatment was 2.33 and we indicated this as (moderately to much improved) to indicate that it fell somewhere in between and closer to the ‘much improved’.

Improvement in ADHD symptoms would not be measured by CGI-S, but rather the symptom scale. One could say symptom severity here and be correct (line 181).

This has been changed to symptom severity.

I am confused about the authors’ statement regarding baseline CGI-I scores across the groups being comparable. CGI-I is a measure of improvement, and would not be relevant here. CGI-S would be a more appropriate measure to compare baseline severity across groups. (line 190)

The reason we think CGI-I scores are appropriate here is because CGI-I scores did change from placebo week, which we would like to highlight

In line 201 I would say that the CGI ratings reflected improved in symptom severity rather than symptoms, since symptoms are measured using a symptom scale such as the Vanderbilt.

This has been changed to symptom severity.

Reviewer #3 (Remarks to the Author):

The authors have sufficiently addressed my comments.

Reviewer #4 (Remarks to the Author):

Reviewer #4, additional comments:

The Authors have responded to all of my comments.

Q1-3 and Q4: As suggested, the authors have rephrased some sentences and corrected a few errors in the original text.

Still, some remaining inaccuracies should be amended.

Without any experimental support, the word «disruptive» to classify the mutations is misleading and should be removed from the abstract and introduction.

The word “disruptive” has been removed from the abstract, introduction and discussion.

Q4: The response to Q4 is still incomplete:

Could the clinical response simply be explained by a placebo effect or ascertainment bias in this open label study?

In the last paragraph of the results we note that a placebo effect is unlikely given that actigraphy data (not influenced by rater bias) showed a reduction by week 5 compared to placebo week.

Ascertainment bias could be playing a role and in the last paragraph of the discussion we include that recruitment was from a single-tertiary care site.

Please explain this more explicitly.

Q5: As the genetic evidence supporting an altered glutamate signaling in ADHD is weak and controversial, the authors should be more cautious in their discussion of this mechanism.

We have revised the following in the discussion section of the manuscript to reflect that alternate mechanisms may explain the therapeutic benefit of Fasoracetam: We recognize that alternative mechanisms explaining the therapeutic efficacy of Fasoracetam may exist. For example, Fasoracetam also affects the cholinergic pathways and thus may also prove to be effective in those ADHD subjects with variants in that pathway. It's also possible that activating NMDA receptors may compensate for dysfunctional cholinergic neurotransmission.³⁴ Further studies are needed to examine the molecular and neurobiologic basis for our observations.

They correctly state that GWAS is not capturing rare events. However, if this signaling system is altered in such a large fraction of ADHD cases, it would not be unexpected to detect signals also in GWAS.

We appreciate the reviewer's comments. This is rather a complex issue but data from others and our own analysis have shown that rare CNVs are poorly captured by common SNPs such as is done with tag-SNPs, for example:

<https://www.ncbi.nlm.nih.gov/pmc/articles/PMC4381751/>.

Even uncommon CNVs (1-5% MAF) may be difficult to tag, for example, as the Wellcome Trust reported that for CNVs with MAF < 5%, 22% have $r^2 > 0.8$ with at least one SNP

(<https://www.ncbi.nlm.nih.gov/pmc/articles/PMC2892339/>). Consequently, we would not expect that classical GWAS approaches would be able to detect these associations, particularly given the heterogeneous disease that is ADHD and that only a few patients may have deleterious structural variants that affect the same genetic locus (individual variants are too rare and have too few subjects at each locus to be able to captured by common variants) (<http://journals.plos.org/plosone/article?id=10.1371/journal.pone.0076295>).

Although possibly captured by a network analysis, most of the genes listed in Table 1A are not GRMs. Many are encoding enzymes involved in diverse biological pathways without any obvious relationship with glutamatergic signaling. Consider relabeling this table.

Table 1a is now Table 1 and is renamed to "Tier-1 mGluR Impacting Gene Network Variants". As explained these are mGluR signaling genes or their interactors that have the potential to impact or modify mGluR signaling.

Likewise, the term "mutations in genes within the mGluR network" should be reserved for genes directly related to glutamate signaling.

This has been changed to "mutations in genes impacting the mGluR network"